# Transient Liquid Phase Diffusion Bonding of Ni_3_Al Superalloy with Low-Boron Nickel-Base Powder Interlayer

**DOI:** 10.3390/ma16072554

**Published:** 2023-03-23

**Authors:** Zhifeng Wen, Qi Li, Fengmei Liu, Yong Dong, Yupeng Zhang, Wei Hu, Likun Li, Haitao Gao

**Affiliations:** 1School of Materials and Energy, Guangdong University of Technology, Guangzhou 510006, China; 2China-Ukraine Institute of Welding, Guangdong Academy of Sciences, Guangdong Provincial Key Laboratory of Advanced Welding Technology, Guangzhou 510650, China; 3Songshan Lake Materials Laboratory, Dongguan 523429, China

**Keywords:** TLP diffusion bonding, interlayer, superalloy

## Abstract

As a technology for micro-deformed solid-phase connection, transient liquid phase (TLP) diffusion bonding plays a key role in the manufacture of heating components of aero engines. However, the harmful brittle phase and high hardness limit the application of TLP diffusion bonding in nickel-based superalloys. In this paper, a new strategy in which a low-boron and high-titanium interlayer can restrain the brittle phase and reduce the hardness of the TLP-diffusion-bonded joint is proposed. With this strategy, the Ni_3_Al joint can achieve a high strength of 860.84 ± 26.9 MPa under conditions of 1250 °C, 6 h and 5 MPa. The microhardness results show that the average microhardness of the joint area is 420.33 ± 3.15 HV and is only 4.3% higher than that of the Ni_3_Al base material, which proves that this strategy can effectively inhibit the formation of the harmful brittle phase in the joint area. The results of EBSD show that 7.7% of the twin boundaries exist in the isothermal solidification zone, and only small amounts of secondary precipitates are observed at the grain boundaries in the joint, which indicates that twin boundaries may play a dominant role in crack initiation. This study provides a feasible avenue to suppress the brittle phase in TLP-diffusion-bonded joints.

## 1. Introduction

Nickel-based superalloys (Ni_0.25_Al_0.75_, NiAl, Ni_0.75_Al_0.25_, etc.) are key materials in the manufacture of advanced aero engines because of their excellent properties, such as resistance to high-temperature deformation, oxidation and corrosion, especially their ability to work at temperatures as high as 85% to 90% of the melting point [1,2,3,4]. Ni_3_Al superalloy has a melting point temperature as high as 1395 °C and excellent yield properties in the middle-temperature range of 600–800 °C due to its new two-phase (γ′-γ) microstructure; therefore, it has received significant research attention and has become one of the most promising candidate materials for blades and impellers of the new-generation aero engine [5,6].

Since Ni_3_Al alloy is commonly used in heating parts with complex structures in aero engines, it is very important to study a suitable bonding method for the manufacture of complex Ni_3_Al alloy parts [7,8]. TLP diffusion bonding is an optimization technology for achieving high strength in Ni_3_Al alloy joints, which is usually performed at high temperatures to ensure sufficient solid-phase diffusion between the bonding interface elements with the base material [9,10]. In the process of TLP diffusion bonding, the interlayer, whose melting point is lower than that of the base material, will first melt and fill the gap between the base material and the interlayer. Melting point depressants (MPDs) such as B, Si and Hf will diffuse from the liquid layer to the junction between the liquid phase and the base material, thus reducing the melting point of the base material at the edge of the liquid phase, resulting in the melting of the base material in this area. This process is essential for the formation of high-quality TLP-diffusion-bonded joints [11,12,13,14,15]. Yingjun Jiao et al. used Ni-Cr-B as the interlayer material for the TLP diffusion bonding of Inconel625 and Mar-M247 superalloy. However, the high levels of borides that precipitate in the joint lead to higher hardness and become the preferred source of cracks, which leads to the deterioration of the mechanical properties of the joint [16]. Zhun Cheng et al. brazed K417G alloy with boron-containing nickel-base alloy powder as the repair filler and found that 2.44% of dispersed M_3_B_2_ boride precipitates were found in the joint [17]. However, the MPDs Si and B in the interlayer can significantly increase the hardness of the joint and lead to a decrease in mechanical properties, which is attributed to the formation of the brittle phase induced by the segregation of silicon and boron [18,19]. Granular and acicular silicides or borides are harmful to the tensile strength and ductility of joints at room or higher temperatures [20,21,22,23].

In order to solve this intractable problem, researchers have made significant efforts. One strategy is to use a pure Ni interlayer without MPDs. The pure Ni interlayer disappears completely after the mutual diffusion between Ni and the base material during long-time heating. However, the performance of the joint is still inferior to that of the base material due to the presence of a non-uniform microstructure diffusion zone at the interface [19,24]. Yu Peng et al. welded single-crystal Ni_3_Al using 4 mm pure nickel foil as the interlayer and obtained a shear strength of 727 MPa at 1170 °C, 55 MPa and 6 h. However, this high pressure can lead to large deformations [25]. Lin Yuan et al. adopted a new 70Mn25Ni5Cr interlayer to perform TLP diffusion bonding on Ni_3_Al and found that high residual MC carbide content in the joint could induce stress concentration and reduce the ductility of the joint. The joint elongation was 4.4% at the highest point, only about 40% of BM [26]. Another strategy is to control the process parameters of diffusion bonding to suppress the influence of borides. The segregation and formation of silicide in the joint are inhibited by reducing the gap size between the bonded substrates and completing isothermal solidification in the appropriate time [27,28]. Hamid Tazikeh et al. used amorphous Ni-Cr-Fe-Si-B filler alloy to conduct vacuum TLP connection for Inconel 939 superalloy. It was found that there were cubic-shaped, needle-like and grain-boundary-shaped borides in the diffusion-affected zone, and the connection temperature has a direct relationship with the hardness of the isothermal solidification zone [29,30]. However, this method is limited by the processing technology. Due to the complex elemental composition of nickel-based superalloys, it is still a challenge to restrain the formation of silicides or borides in the joint during TLP diffusion bonding and to make the joint hardness close to that of the base material.

In the TLP diffusion bonding of Ni_3_Al superalloy, the commonly used interlayers are BNi-2 (boron 3.2 wt.%), MBF-30 (boron 3.2 wt.%), NB-150 (boron 3.5 wt.%), etc. The content of the B element in all of these interlayers exceeds 3.2 wt.%. Here, we design an interlayer with low B (1.27 wt.%) and high Ti to meet this challenge. It is well known that the melting point temperature of the alloy drops sharply from 1670 °C to 942 °C as the Ti content increases from 0 to 23% in the Ti-Ni phase diagram [31]. Thus, it is possible to inhibit the formation of borides and silicates in joints by adding Ti and reducing B and Si MPDs in the interlayer. Enlightened by the above findings, we prepared a diffusion bonding interlayer with low B and high Ti and studied Ni_3_Al-based alloy TLP-diffusion-bonded joints with this interlayer. A joint with a high strength of 860.84 MPa was obtained by TLP diffusion bonding, which is an intuitive parameter for judging the quality of the TLP-diffusion-bonded joint [6,32,33]. Unexpectedly, the average microhardness in the joint area was 420.33 HV, only 4.3% higher than that of the Ni_3_Al base material. This strategy effectively inhibits the formation of the harmful brittle phase in the joint area. In order to understand the properties of the interlayer with low B and high Ti in TLP diffusion bonding, systematic experiments have been performed.

## 2. Materials and Methods

### 2.1. Material Preparation

Ni_3_Al superalloy (IC10) columnar bars with a size of Φ 16 mm × 200 mm and orientation of [001] were used as the base material (Institute of Metals, Chinese Academy of Sciences, Shenyang, China). The chemical compositions of the Ni_3_Al superalloy and the interlayer alloy powder are shown in Table 1. The preparation method of the interlayer powder are as follows: Ni, Cr, Ta, Al, W, Mo, Ti, Nb and Re metal particles with 99.9% purity and crystal B (Zhongnuo Advanced Material (Beijing) Technology, Beijing, China) undergo mixed melting in a vacuum melting furnace (VCF, Shenyang Vacuum Technology Institute, Shenyang, China) at 1450 °C, then the alloy ingot with uniform composition is obtained after five cycles of vacuum melting, and lastly, the interlayer alloy powders are formed by water atomization under an argon atmosphere (VCF, Shenyang Vacuum Technology Institute, Shenyang, China). As shown in Figure 1, more than 80% of the particles range from 22.1 µm to 160.6 µm in diameter, and the largest proportion of particle diameters is 91.2 µm. The density of the powder is 7.98 g/cm^3^. According to the density formula, the powder mass required to completely cover the bonding surface is calculated to be about 0.022 g based on an interlayer thickness of 40 μm. In order to ensure the precision of the powder weight in the interlayer, an electronic balance with a precision of 0.001 g was used for weighing. Figure 1c shows the DTA curve of the interlayer powder. The melting point of interlayer powder is 1112.5 °C, and that of the Ni_3_Al base material is 1390 °C, 277.5 °C lower than that of the Ni_3_Al base material. Consequently, the suitable TLP diffusion bonding temperature range using a low-B nickel-base power interlayer is about 1160~1300 °C.

### 2.2. Transient Liquid Phase Diffusion Bonding

The samples were cut into sticks of Φ 16 mm × 30 mm via wire electrical discharge machining along the [100] direction, and the schematic diagrams of the diffusion bonding fixture are shown in Figure 2. Four samples can be installed in the fixture and positioned with a positioning circle (lower die) and an alumina ceramic gasket (upper die). Before diffusion bonding, the surface of the Ni_3_Al substrate was treated with 200–2000# SiC sandpaper to remove oxides and contaminants. Then, the treated samples were ultrasonically cleaned in alcohol for 10 min. TLP diffusion bonding was carried out with a vacuum diffusion bonding furnace (Central Vacuum Industries, Workhorse II, Nashua, NH, USA). The vacuum degree before heating was higher than 5 × 10^−4^ Pa. The TLP diffusion bonding was performed at 1250 °C for 6 h under a pressure of 5 MPa. The heating curve and pressure curve are shown in Figure 2c.

### 2.3. Characterization Methods

The cross-section of the TLP diffusion joint was mechanically polished and chemically etched in a solution of glycerin (5 mL) + HF (10 mL) + HNO_3_ (5 mL) for 30–35 s. The microstructure and chemical composition of TLP-diffusion-bonded joints were examined with a field-emission scanning electron microscope (SEM, Nova Nano, SEM430) equipped with an energy-dispersive X-ray spectroscopy instrument (EDS, Noran System 7, ThermoFisher Scientific, Waltham, MA, USA). The density of the powder was measured with a densitometer (HR-D1, Dandong Hengrui Instrument Co., Liaoning, China). The thermal analysis of the interlayer powder was carried out with a differential thermal analyzer (Netzsch PC409, NETZSCH GmbH & Co., Selb, Germany). A transmission electron microscope (TEM, JEOL-2100F, JEOL, Tokyo, Japan) equipped with a high-angle annular dark-field (HAADF) detector and electronic back-scatterer detection (EBSD, Oxford, Edax, Oxford, UK) instrument were used to analyze the fine microstructure, and the phase composition of the joint was tested using an X-ray diffractometer (XRD, RIGAKU Smartlab 9 kW, Smartlab, Tokyo, Japan). The EBSD samples were prepared by OPS polishing. An X-ray photoelectron spectrometer (XPS, Thermo Fisher Nexsa, Waltham, MA, USA) was used to record the XPS spectra of a typical sample. The mechanical properties of the sample were tested with a universal testing machine (MTS, CMT5105, Belmont, MS, USA). The elasticity modulus was measured with a nanoindentation tester (Anton Paar, UNHT3, Graz, Austria) with a load of 30 mN and a loading duration of 55 s, and the microhardness was measured with a Vickers microhardness tester (Wilson VH1202, Wilson, Fort Worth, TX, USA) with a load of 300 g and a loading duration of 10 s.

## 3. Results and Discussion

### 3.1. The Microstructure of Ni_3_Al TLP-Diffusion-Bonded Joints Using Low-B Interlayer

Figure 3 shows the scanning electron micrograph of the cross-section of a TLP-diffusion-bonded joint using a low-B interlayer. The joint is composed of an isothermal solidification zone (ISZ), a diffusion-affected zone (DAZ) and the Ni_3_Al substrate, exhibiting the typical microstructure characteristics of the TLP-diffusion-bonded joint. The width of the ISZ is about 50 µm. It is important to obtain the ideal microstructure of the transition area between the joint and substrate for a high-performance joint. As shown in Figure 3b, the DAZ, as the transition region between the joint and the Ni_3_Al substrate, presents a uniform and fine mesh structure with a diameter of less than 1 μm, similar to the microstructure of the Ni_3_Al substrate, indicating that the joint grows epitaxially with the substrate and has a good bonding interface. Compared with the Ni_3_Al matrix, the diameter of the unit structure in the DAZ is smaller, which is beneficial for enhancing the joint strength.

The SEM-EDS results of the DAZ are shown in Table 2. The dark region (#1) and light region (#2) are the γ′ phase (Ni_3_Al) and γ phase (Ni), respectively. In addition, Ti is mainly distributed in the γ′ phase, because Ti tends to replace Al in Ni_3_Al [34,35,36]. Since the Ni_3_Al substrate does not contain the Ti element, the concentration of the Ti element in the interlayer is significantly higher than that in the Ni_3_Al substrate. Under the influence of a chemical concentration gradient, the Ti element diffuses from the interlayer to the Ni_3_Al substrate, resulting in a decrease in Ti content in the joint. The Cr and Co elements tend to concentrate in the γ phase, while Ta mainly concentrates in the γ′ phase. The behavior of Cr, Co and Ta can be attributed to the partitioning behavior of these alloying elements between the two phases [37,38]. The EDS results show that the precipitates in the DAZ (#3) are Hf-rich carbides composed of Co, Ni, Hf and C elements. Obviously, the Hf-rich precipitates are caused by the diffusion of Hf from the Ni_3_Al substrate to the joint. The precipitates (#4) in the ISZ are intermetallic compounds mainly composed of W, Cr and Ni. Only very small amounts of precipitated carbides and intermetallic compounds are observed in the DAZ and ISZ, and almost no precipitated boride phase is observed, indicating that the low-boron interlayer can effectively inhibit hard boride phase formation.

In order to observe the microstructure of the joint more clearly, TEM images of the TLP diffusion bonding joint region are shown in Figure 4. The joint microstructure is mainly composed of an elliptical region and an ellipse-surrounding region, which form grids. According to the bright-field TEM image (Figure 4a–d), the diameters of the elliptical regions range from 412 nm to 1226 nm, and the average diameter is about 721 nm. According to the TEM-EDS results (Table 3), the elliptical region and the ellipse-surrounding region consist of the γ′ phase and γ phase, respectively. The Ti content of the γ′ phase in the joint is only 0.11%, because Ti has a high solid-phase diffusion coefficient and diffuses from the joint to the Ni_3_Al substrate at high temperatures, which is consistent with the results of SEM-EDS in Table 2. Studies have shown that when the Ti content exceeds 1.7%, the γ′ phase changes from an ellipsoidal phase to a cubic phase [37], which explains why the γ′ phase in the joint maintains an elliptical structure. In addition, Ti can reduce the interfacial energy between the γ′ phase and γ phase, thus increasing the driving force for the growth of the γ′ phase. The increase in Ti content in the joint will lead to an increase in the microstructure size of the γ′ phase. Therefore, maintaining low Ti content in the joint is the key to forming fine γ′ phases.

It is generally believed that the γ′ phase has enhanced directivity, and the ratio of the γ′ phase to the γ phase directly affects the mechanical properties of non-equilibrium materials composed of γ′ + γ. According to the TEM image (Figure 4a–d), the area ratio of the γ′ phase to the γ phase in the joint is 49.3:50.7. Increasing the proportion of the γ′ phase can improve the strength, but the toughness decreases. While increasing the proportion of the γ phase can increase the toughness, the strength decreases. Therefore, it is important to maintain the proper area ratio of the γ′ phase to the γ phase in the joint. The proper volume fraction of the γ′ phase guarantees the high stability of Ni_3_Al at high temperatures [26]. James et al. found that lattice mismatches were greatly mitigated when the volume fraction of the γ′ phase was reduced to ~50% during the creep process, thus enhancing the resistance to creep [39]. Therefore, the ratio of the γ′ phase to the γ phase in the joint is conducive to obtaining good creep resistance at high temperatures.

Figure 4e shows the selected area diffraction pattern of the joint region, which is indexed from Figure 4a. There are two groups of diffraction spots: the brighter spots are the γ′ phase, and the darker spots are the γ phase. Figure 4f and Figure 4a–d,g show that there are a large number of parallel dislocation lines in both the γ′ phase and γ phase of the joint. The length of the dislocation lines is about 50–300 nm, and the dislocation line density is 1.84 × 10^14^ m^−2^ according to the TEM bright-field image statistics. Dislocation lines will entangle with each other by slipping and climbing, which requires more energy to initiate the dislocation line motion, resulting in a dislocation-induced strengthening effect on the mechanical strength of the joint. Studies have shown that pre-existing dislocations in Ni_3_Al superalloy can significantly increase the creep latency under low-stress creep conditions and provide Ni_3_Al superalloy with excellent creep strength [39].

Figure 5 shows EBSD images of the cross-section of the TLP-diffusion-bonded joint using the low-B interlayer. The ISZ is mainly composed of randomly oriented equiaxed grains, and the width of the ISZ is about 50–60 μm, which is consistent with the results in Figure 3. Since grain boundary failure is one of the main failure forms of superalloys, the grain boundary information for the TLP-diffusion-bonded joint was obtained and is given in Figure 5b. The ISZ is mainly composed of large-angle grain boundaries (blue lines) at 15~180°, which is attributed to heterogeneous nucleation during the solidification of the TLP diffusion bonding process. Thermal cracks caused by liquid film, stress concentration and Re-rich precipitates at grain boundaries are common serious defects in the solidification process of nickel-based superalloys. Thermal crack initiation tends to occur at small grain boundaries, because the small-angle grain boundary has a higher degree of undercooling during solidification [40]. The ISZ of the joint is mainly composed of large-angle grain boundaries, which can effectively reduce the tendency for thermal crack formation during the TLP diffusion bonding process.

In addition to thermal cracks, fatigue cracking is also an important cause of superalloy failure. Non-metallic precipitates and twin boundaries are preferred initiation locations of fatigue cracks in polycrystalline nickel-based superalloys. Therefore, the secondary precipitates and twin boundaries at the grain boundaries were studied. The twin boundaries of nickel-based superalloys are prone to failure in harsh environments, because the critical stress of dislocation slip is lower at the twin boundary than at the common grain boundary and within the grain [41]. As shown by small arrows in Figure 5b, there is a small number of twin grain boundaries in the ISZ zone, accounting for about 7.7% of the grain boundaries, while only very small amounts of secondary precipitates are observed at the grain boundaries of the joint. There is competition between non-metallic precipitates and twin boundaries for the initiation of fatigue cracks [42]. Previous studies have found that the interaction between dislocations and grain boundaries has an important effect on grain boundary crack initiation. When the dislocation is parallel to the twin boundary, it is difficult for the dislocation to move through the twin boundary, resulting in stress concentration at the twin boundary. This leads to crack initiation at the twin boundary [43,44]. The number of twin grain boundaries is significantly higher than that of non-metallic precipitates at grain boundaries in the TLP-diffusion-bonded joints using the low-B interlayer, so twin boundaries may play a dominant role in crack initiation.

Figure 6 shows the inverse pole diagram of the TLP-diffusion-bonded joint using a low-B interlayer. The strengths of [001], [100] and [010] are 21.385, 7.704 and 2.776, respectively, indicating that [001] is the preferred orientation.

### 3.2. The Phases of Ni_3_Al TLP-Diffusion-Bonded Joints Using Low-B Interlayer

The GIXRD spectrum of the joint region is shown in Figure 7. The peak at 75° corresponds to Ni_3_Al, while the peaks at 43° and 91° correspond to Ni and Ni_3_Al, which are the main phases of the joint. The diffraction peaks at 39.5° and 68° correspond to HfC, and the diffraction peak at 96° corresponds to TaC. The peak intensity of the two components is very weak, indicating that the carbide content is very low. The peaks at 51° and 57° correspond to boride (CrB, Cr_1.8_W_3.2_B_3_), indicating that B in the joint tends to bind Cr and W.

In order to further study the phases in the joints, the XPS spectra of Ni 2p, Cr 2p and Ta 4f in the joints were obtained and are shown in Figure 8. The binding energy is corrected with reference to the C 1s peak at 284.6 eV. Figure 8a shows that there are three main peaks in the Ni 2p orbital: the peak at 852.8 ev corresponds to Ni_3_Al (CAS Registry No. 12063-96-6), and the peaks at 874.1 ev and 879.9 ev correspond to Ni (CAS Registry No. 7440-02-0). The peak at 855.6 ev is Ni-O formed by slight oxidation. As shown in Figure 8b, the peaks at 574.1 ev, 577.1 ev and 583.1 ev in the Cr 2p orbital correspond to Cr_7_C_3_ and CrBO_3_ (CAS Registry Nos. 12007168, 12075-40-0), and the peak at 586.9 ev is caused by Cr-Ni-O. Obviously, Cr easily combines with C and B to form carbides and borides in addition to Ni to form intermetallic compounds. As seen in Figure 8c, the peaks at 22.9 ev and 24.9 ev in the Ta 4f orbital are due to TaC_0.95_ and Ta_2_C, respectively (CAS Registry No. 12070-06-3), which shows that Ta in the joint readily generates the hard carbide phase.

### 3.3. The Mechanical Properties of Ni_3_Al TLP-Diffusion-Bonded Joints Using Low-B Interlayer

Figure 9 demonstrates the tensile strength curve of the TLP-diffusion-bonded joint. The average tensile strength of the joint is 860.84 ± 26.9 MPa, which indicates that a joint with excellent mechanical properties can be formed by TLP diffusion bonding with the low-B and high-Ti interlayer strategy. The reasons for the high strength of the joint may be as follows: (1) The fine grid structure of the γ′ phase and γ phase is formed at the DAZ zone of the joint area, which is conducive to the improvement in the joint strength. (2) A large number of dislocation lines formed in the joint, resulting in dislocation strengthening. (3) The low-B interlayer inhibits the formation of the hard and harmful boride phase in the joint.

Figure 10 shows the microhardness (from the Vickers microhardness test) and elastic modulus (from the nanoindentation test) results of the cross-section of the TLP-diffusion-bonded joint. The average microhardness of the Ni_3_Al substrate and joint area is 403.71 ± 4.42 HV and 420.33 ± 3.15 HV, respectively. The average microhardness of the latter is only 4.3% higher than that of the former, indicating that only a small amount of the harmful brittle phase is produced in the joint area. In the TLP diffusion bonding process of Ni_3_Al superalloy, the interlayer first liquefies to fill the gaps between the base material and the interlayer, so the melting point of the interlayer is required to be lower than that of the base material. The melting point of the interlayer is often reduced by adding the boron element. However, the addition of boron is a double-edged sword. Although the boron element reduces the melting point of the interlayer, the boron element in the interlayer tends to form a hard and brittle harmful phase of borides. The hardness test proves that the formation of the hard phase in the joint is effectively inhibited. The average elastic moduli of the Ni_3_Al base metal and the joint area are 210.86 ± 6.54 GPa and 228.66 ± 5.09 GPa, respectively. The average elastic modulus of the joint area increased by 8.4% compared with that of the Ni_3_Al base metal. The elastic modulus showed the same trend as the microhardness, further verifying the microhardness test results. The increase in the elastic modulus is higher than that in microhardness, which may be related to the strengthening effect caused by the large number of dislocation lines in the joint.

### 3.4. The Fracture Morphology of Ni_3_Al TLP-Diffusion-Bonded Joints Using Low-B Interlayer

Figure 11 shows the fracture morphologies of the TLP-diffusion-bonded joint. The fracture of the tensile specimen is located in the diffusion-bonded joint. The fracture morphology is uniform, indicating that the interface of the joint contributes uniform mechanical properties. As shown in Figure 11b, the fracture is a mixed fracture mode, and there are densely distributed small dimples with diameters of 0.5–2 µm in the fracture area, which contribute to the joint toughness.

## 4. Conclusions

In summary, aiming to address the difficulty in Ni_3_Al TLP-diffusion-bonded joints caused by borides, which increase the hardness and decrease the mechanical properties of the joints, a low-boron nickel-base powder interlayer is proposed to reduce the hard phase of TLP-diffusion-bonded joints, and high-strength Ni_3_Al superalloy joints are formed by transient liquid phase diffusion welding. The conclusions can be summarized as follows: A joint with a high strength of 860.84 ± 26.9 MPa was obtained at a diffusion welding temperature of 1250 °C, a holding time of 6 h and a welding pressure of 5 MPa by using a low-B interlayer. The microhardness results show that the average microhardness of the joint area is 420.33 ± 3.15 HV, which is only 4.3% higher than that of the Ni_3_Al base material. It is proved that the low-boron nickel-base interlayer strategy effectively inhibits the formation of the harmful brittle phase in the joint region. The average elastic modulus of the joint region is 228.66 ± 5.09 GPa, which is 8.4% higher than that of the Ni_3_Al base material. The elastic modulus and microhardness show the same change trend, further validating the conclusion of the hardness test. The SEM results show that the microstructure of the DAZ zone in the joint transition region is uniform with a diameter of less than 1 µm and has a similar structure to the Ni_3_Al-based structure. This fine structure is conducive to strengthening the joint strength and also indicates that the joint and Ni_3_Al base material have a good bonding interface. The results of EBSD show that 7.7% of twin boundaries exist in the isothermal solidification zone, and only small amounts of secondary precipitates are observed at the grain boundaries in the joint, which indicates that the twin boundaries may play a dominant role in crack initiation. In addition, bright-field transmission electron microscopy images show that a large number of dislocation lines are generated in the joint region with a density of 1.84 × 10^14^ m^−2^, indicating that there is significant dislocation strengthening in the joint. This study contributes to the application of diffusion welding in superalloys.

## Figures and Tables

**Figure 1 materials-16-02554-f001:**
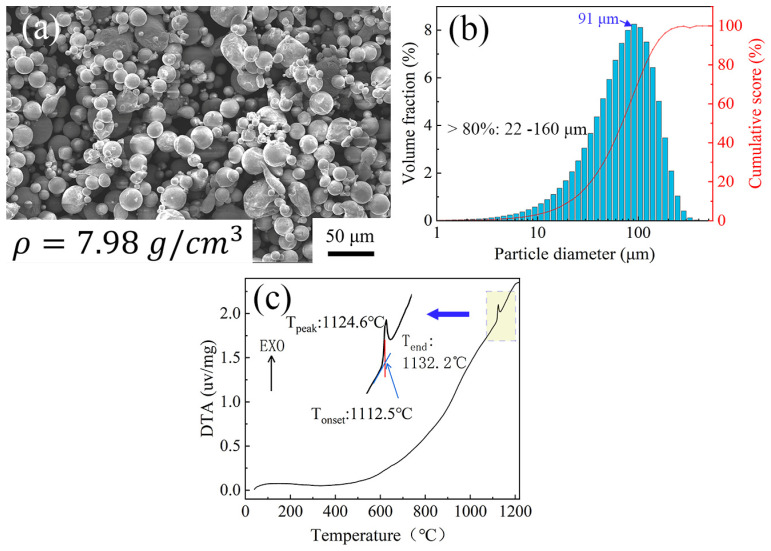
SEM image, particle diameter distribution bar graph and DTA curve of interlayer powder: (**a**) SEM image; (**b**) particle diameter distribution bar graph; (**c**) DTA curve.

**Figure 2 materials-16-02554-f002:**
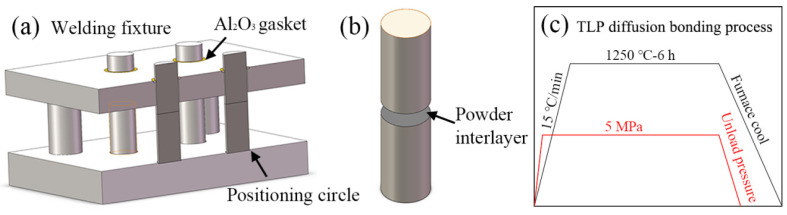
Specimen assembly diagrams and TLP diffusion bonding process: (**a**) welding fixture; (**b**) powder interlayer assembly diagram; (**c**) TLP diffusion bonding process.

**Figure 3 materials-16-02554-f003:**
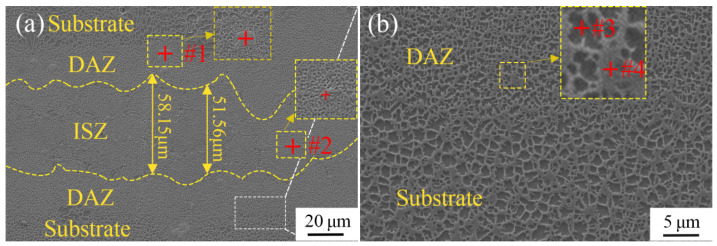
SEM images of cross-section of TLP-diffusion-bonded joint using low-B interlayer. (**a**) TLP joint bonded at 1250 °C for 6 h; (**b**) Expanded region in (**a**).

**Figure 4 materials-16-02554-f004:**
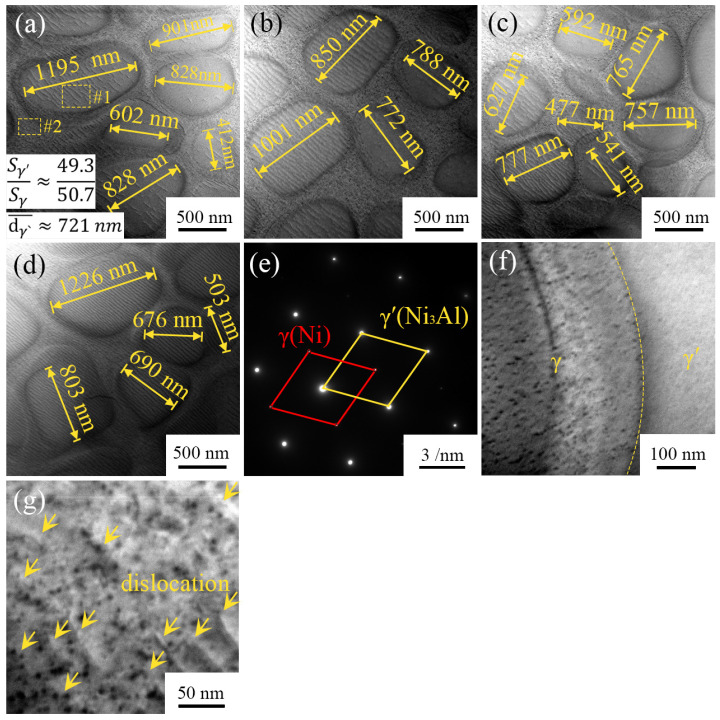
TEM images of TLP-diffusion-bonded joint: (**a**–**d**,**f**,**g**) bright-field TEM image and (**e**) selected area diffraction indexed from (**a**).

**Figure 5 materials-16-02554-f005:**
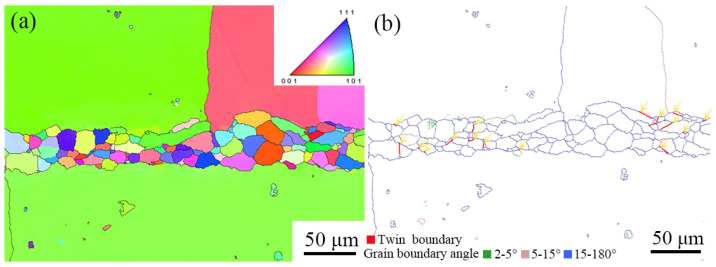
EBSD images of TLP-diffusion-bonded joint: (**a**) IPF image; (**b**) grain boundary angle distribution.

**Figure 6 materials-16-02554-f006:**
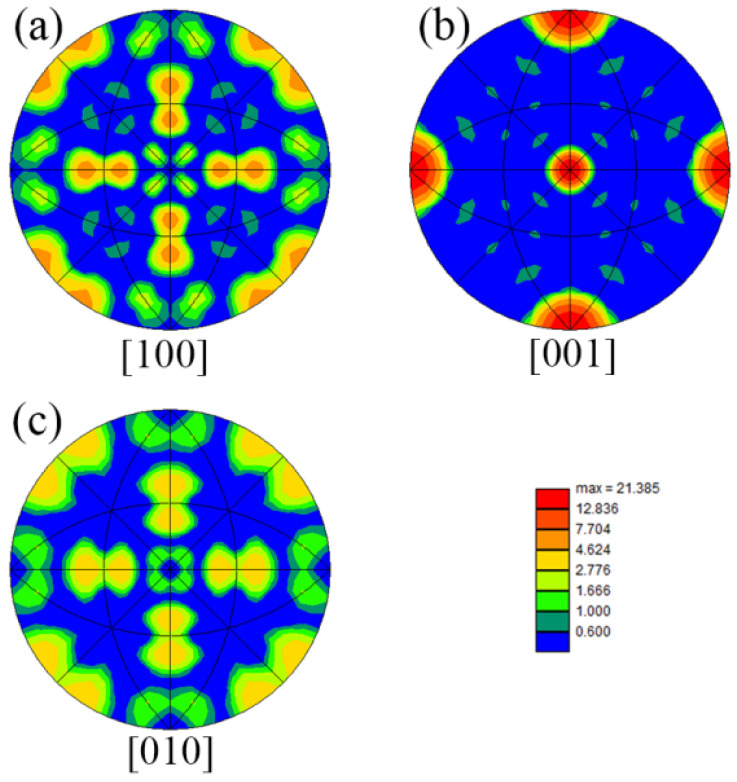
Inverse polar diagram of TLP-diffusion-bonded joint using low-B interlayer. (**a**) [100] orientation; (**b**) [001] orientation; (**c**) [010] orientation.

**Figure 7 materials-16-02554-f007:**
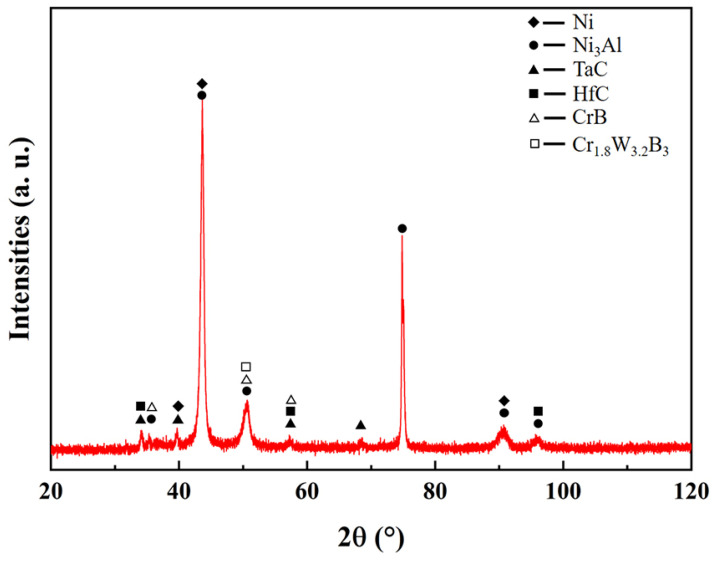
XRD pattern of TLP-diffusion-bonded joint using low-B interlayer XRD.

**Figure 8 materials-16-02554-f008:**
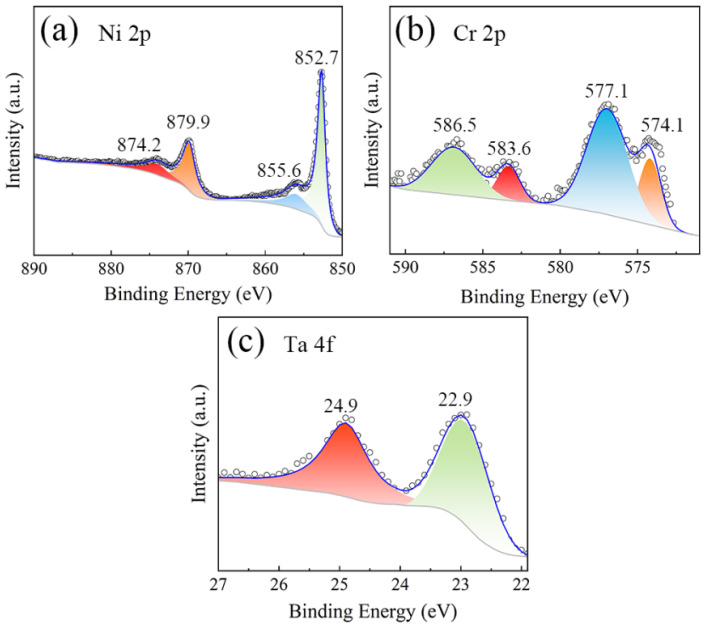
XPS pattern of TLP-diffusion-bonded joint using low-B interlayer. (**a**) Ni 2p; (**b**) Cr 2p; (**c**) Ta 4f.

**Figure 9 materials-16-02554-f009:**
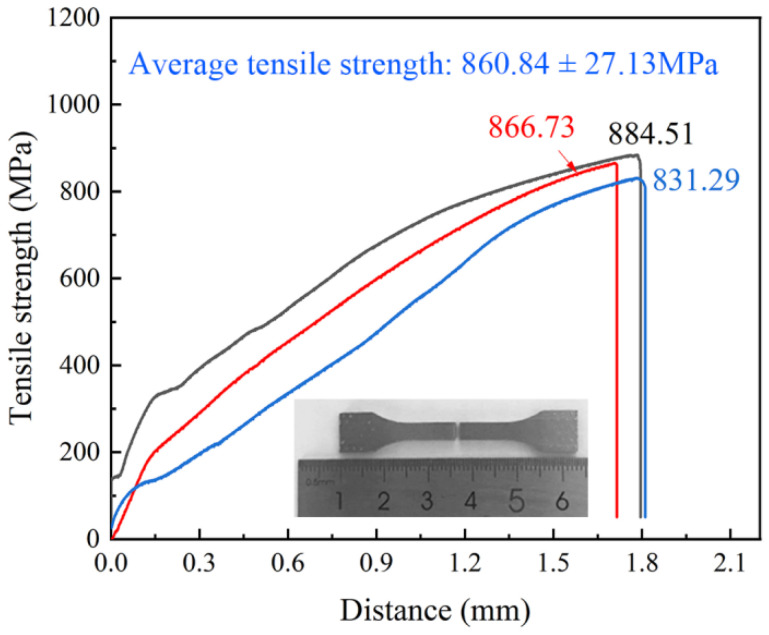
Tensile strength of TLP-diffusion-bonded joint using low-B interlayer.

**Figure 10 materials-16-02554-f010:**
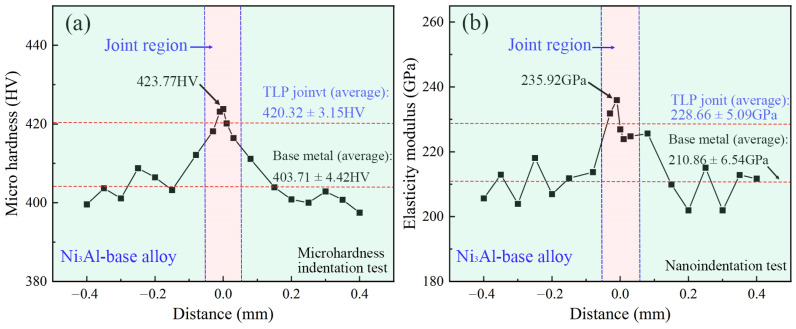
Microhardness curve and elasticity modulus curve of TLP-diffusion-bonded joint using low-B interlayer. (**a**) Micro hardness of TLP joint; (**b**) Elasticity modulus of TLP joint.

**Figure 11 materials-16-02554-f011:**
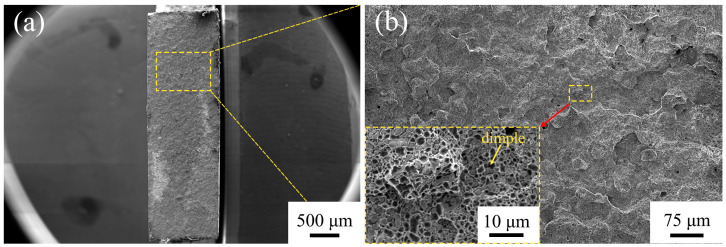
The fracture morphologies of TLP-diffusion-bonded joint using low-B interlayer. (**a**) Macroscopic fracture morphology; (**b**) Expanded region in (**a**).

**Table 1 materials-16-02554-t001:** Chemical compositions of powder interlayer and Ni_3_Al base material (wt.%).

Materials	Co	Cr	Ta	Al	W	Hf	Mo	Ti	Nb	Re	B	C	Ni
Interlayer	7.37	12.61	3.75	3.20	4.64	-	0.96	4.51	0.35	2.45	1.27	1.37	Bal
Ni_3_Al- base alloy	12.15	6.94	6.87	6.02	4.81	1.63	1.22	-	-	-	-	1.21	Bal

**Table 2 materials-16-02554-t002:** SEM-EDS scan results according to Figure 3 (at.%).

Position	C	Al	Ti	Cr	Co	Ni	Mo	Hf	Ta	W
#1	21.25	-	0.27	7.07	11.51	31.73	0.47	24.76	2.94	-
#2	3.84	3.15	-	22.10	2.99	14.57	-	-	2.12	51.23
#3	6.29	11.45	0.38	4.98	8.51	64.58	0.31	0.24	1.65	1.61
#4	6.34	2.81	-	11.54	13.64	64.33	-	-	-	1.34

**Table 3 materials-16-02554-t003:** TEM-EDS scan results according to Figure 4 (at.%).

Area	C	Al	Ti	Cr	Co	Ni	Mo	Hf	Ta	W
#1	4.13	12.17	0.11	3.73	8.74	66.16	1.03	-	1.56	1.37
#2	3.58	3.99	0.06	14.42	15.39	61.19	0.34	-	-	1.03

## Data Availability

The data presented in this study are available on request from the corresponding author after obtaining permission from the authorized person.

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
