# Peer review of "Transient Liquid Phase Diffusion Bonding of Ni3Al Superalloy with Low-Boron Nickel-Base Powder Interlayer"

_materials, 2023, doi:10.3390/ma16072554_

Round 1

Reviewer 1 Report

The main question addressed by the research is , a new strategy that with low boron and high titanium interlayer 15 can restrain the brittle phase and reduce the hardness of TLP diffusion bonding.   The topic is original and relevant in the field and is important in the related area.
Methodology is adequate.   The conclusions are consistent with the evidence and arguments presented and do they address the main question posed.   The  references are appropriate.   Figs are presented with quality.

Author Response

Thank you for your review and approval of this manuscript.

Reviewer 2 Report

The content of the paper titled: “Transient liquid phase diffusion bonding of Ni3Al superalloy with low boron nickel-base powder interlayer” has successfully studied the liquid phase diffusion bonding of Ni3Al superalloy with the interlayer of base nickel powder. The results obtained are very attractive but cannot be published. The author needs to edit and improve to the following comments:

- Check the abbreviations, document symbols in the manuscript and fully add titles, abbreviation labels, document symbols in all parts of the manuscript.

- It is necessary to detail the experimental procedure, name of materials, measuring equipment, manufacturer, accuracy of equipment, place of equipment, ...

- It is necessary to add the work results from 2018 to the present and present the latest results on the content of the manuscript in the introduction and conclusion to highlight the content of the manuscript. Besides, it is possible to change the title of the article content to suit the content.

In addition, the following article can add an interesting part of AlNi phase alloy to study the factors affecting the microstructure, phase transition and crystallization of the material.

+ Factors affecting the structure, phase transition and crystallization process of AlNi nanoparticles

The authors need to answer the following questions:

+ Why use low boron nickel powder to reduce the hard phase of TLP diffusion bonding? what is low boron nickel powder needs a detailed explanation? This concept is very vague and needs to be clarified.

+ Why the bonding of Ni3Al alloy is determined by the diffusion mechanism of the liquid phase?.

+ Why research only with Ni3Al alloy but not with different doped alloys such as Ni0.25Al0.75, NiAl, Ni0.75Al0.25.

+ Why is the author only interested in the high strength junction 860.84 ± 26.9 MPa at 1250 ℃.

+ Why does the author assert that the twin boundary can play a key role in creating cracks? where does this crack come from and what causes it?

- All the above requirements should be fully explained by the author and fully added to the manuscript content to increase the attractiveness and highlight the outstanding issues.

Congratulations to the author on success with this useful work.

Author Response

Thank you for your review and valuable suggestions for the revision of this Manuscript. Please see the attachment.

Reviewer 3 Report

The article is written on a very relevant topic and is undoubtedly recommended for publication. As a technology of microdeformed solid-phase connection, transient liquid diffusion bonding plays a key role in the manufacture of heating components of aero engine.

  However, there are small remarks: on line 288, the word Discussion should be corrected to Conclusions, a lot of results have been obtained and the conclusions should be somewhat expanded.

Author Response

Thank you for your review and valuable suggestions for the revision of this Manuscript. We have corrected the mistakes indicated by the reviewer in detail.

Reviewer 4 Report

Dear Authors,

I find some of your findings interesting albeit I would like to point out the following comments,

1: p5-l154: how come the miniumum and maximum diameter of the eliptical region are found to be in one micrograph, is there no more images taken? could the authors provide more measurements/statistics?

2: Fig 4: Daverage is reported to be 762 out of how many measurements? Fig4b the SAD should be indexed, the lines drawn on the DP are not clear as to how this would be gamma and gamma prime, aren’t they supposed to be the other way around?! Fig 4d: in order to show dislocations a two beam image is required to show the presence of dislocations, why are almost all of them oriented in one direction?

3: p10: discussionàSummary and Conclusion

It is not clear what the authors intent to say in this sentence” The results of EBSD show that 304 7.7% twin boundary exists in isothermal solidification zone, and only a small amount of 305 secondary precipitates are observed at the grain boundary in joint, which indicates that 306 the twin boundaries may play a dominant role in crack initiation.

How can the SAD give information on dislocation density? “In addition, TEM results 307 of selected area diffraction show that a large number of dislocation lines are generated in 308 the joint redion (should be corrected to region) with the density

4: p5: Here the dislocation density is reported to be of the order of 10^8 m-2 which is very low. How is this density measured why is it too low when looking at the image one can estimate it to be higher.

The dislocation density in the way it is calculated and reported could be affected by the thickness, this method is not suitable for reporting evaluations on dislocations.

5: fig 7: the XRD peaks should be indexed.

Author Response

(The authors gave the same response as above.)

Round 2

Reviewer 2 Report

Ask the author to review each question and answer each problem raised, and add to the content of the manuscript, where changes should be highlighted in red for easy tracking. In addition, the content of the answer must match the content of the manuscript because this is the last time for the author to edit.

Round 3

Reviewer 2 Report

Accept online publication, after the author has checked the entire English style and grammar for the last time without having to send it back to the reviewer.